# Exponential increase in mortality with age is a generic property of a simple model system of damage accumulation and death

Anders Ledberg *

Department of Public Health Sciences, Stockholm University, Stockholm, Sweden

* anders.ledberg@su.se, anders.ledberg@gmail.com

## Abstract

The risk of dying increases exponentially with age, in humans as well as in many other species. This increase is often attributed to the "accumulation of damage" known to occur in many biological structures and systems. The aim of this paper is to describe a generic model of damage accumulation and death in which mortality increases exponentially with age. The damage-accumulation process is modeled by a stochastic process know as a queue, and risk of dying is a function of the accumulated damage, i.e., length of the queue. The model has four parameters and the main characteristics of the model are: (i) damage occurs at random times with a constant high rate; (ii) the damage is repaired at a limited rate, and consequently damage can accumulate; (iii) the efficiency of the repair mechanism decays linearly with age; (iv) the risk of dying is a function of the accumulated damage. Using standard results from the mathematical theory of queues it is shown that there is an exponential dependence between risk of dying and age in these models, and that this dependency holds irrespective of how the damage-accumulation process is modeled. Furthermore, the ways in which this exponential dependence is shaped by the model parameters are also independent of the details of the damage accumulation process. These generic features suggest that the model could be useful when interpreting changes in the relation between age and mortality in real data. To exemplify, historical mortality data from Sweden are interpreted in the light of the model. The decrease in mortality seen between cohorts born in 1905, compared to those born in 1885, can be accounted for by higher threshold to damage. This fits well with the many advances made in public health during the 20th century.

## Introduction

In many biological organisms, including humans, mortality rate (force of mortality or hazard rate) is increasing with age [e.g., 1, 2]. This increase is often taken as a defining feature of biological aging, or senescence. A remarkable empirical finding, first described almost 200 years ago, is that the rate of increase in mortality, over a substantial age range, is roughly exponential [3]; see [4] and [5] for review. This exponential relationship is found in different populations

**Funding:** This research was partly financed by a grant from Statens institutionsstyrelse with number 2.6.1-1134-2017. There was no additional external funding received for this study.

**Competing interests:** The authors have declared that no competing interests exist.

of humans as well as in a range of other organisms, including fruit flies and nematodes. Moreover, the exponential dependence between age and mortality rate is seen also under experimental conditions where environmental, and to some extent genetic, forces are kept constant [e.g., 6]. In human populations, the age range over which there is an exponential relationship between age and mortality rate varies. In Swedish data, mortality is an approximately exponential function of age from 55 years of age [7, Ch.6].

A prominent idea, often evoked in explanations of why humans and other organisms age and die, is that damage, that inevitably occur in biological systems, accumulate over time, and the accumulated damage leads to organ failure, and finally death [e.g., 2, 8–11]. The "damage" featuring in these accounts is of many different kinds and occur to a range of different structures, including nuclear and mitochondrial DNA, and proteins [9]. The rate of damage accumulation, and hence of aging, is thought to be a result of the balance between damage and repair. There are a range of repair mechanisms, the most well studied repair damage to the DNA [12, 13]. Many of these repair mechanisms become less efficient with age [12], which implies that the rate of damage accumulation will increase with age.

The aim of this paper is to describe a family of models of damage accumulation and death, where there is an exponential dependence between rate of dying and age, a dependence that can be shaped by a small set of parameters having a clear interpretation. The description of damage accumulation in these models is taken from the mathematical theory of queues, and the process of accumulation is shaped by the balance between the rate of damage and rate of repair. The model systems "die" at a fixed rate when the accumulated damage exceeds a threshold, and if the rate of repair decreases in a roughly linear manner with age, mortality rates can be shown to be exponential functions of age.

In the simplest instantiation of such queuing-based models, there are four parameters that shape the relation between aging and mortality rate. The way in which mortality rate depends on the parameters is shown to be independent of the details of how the system is implemented, suggesting that model systems of this type could be useful in interpreting changes in real mortality data.

The paper is organized as follows. First a model system is studied where damage accumulation is modeled by an M/M/1 queue with time-changing repair (service) rate. This queuing model is well understood, and the mortality rates are shown to be well approximated by an exponential function of age. Second, it is shown that the relation between age and mortality is approximately exponential also when more general queuing models are used to describe the damage accumulation. Moreover, it is shown that these more general models can be parameterized in the same way as the model based on M/M/1-queues, and the effects of the parameters are the same in the two models. In the following sections numerical methods are used to study two cases not covered by the theory. First the case of when rate of damage accumulation exceeds rate of repair is considered. This is followed by a discussion of how heterogeneity can be introduced in these models. Subsequently, historical mortality data from Sweden are interpreted in the light of the damage accumulation models. In the Discussion, these model systems are related to some previously suggested models of similar type. In the S1 Appendix, numerical simulations are used to verify that the theoretical approximations accurately describe the dynamics of real systems, and it is furthermore shown that the assumption of a hard threshold can be relaxed.

## Damage accumulation modeled by an M/M/1 queue

In this section, a well-known model of queuing behavior, the M/M/1 queue, is used to implement a system that ages, and moreover dies with a rate which depends exponentially on age. For an introduction to queuing theory see, for example, [14].

Assume that damage to the system occurs according to a Poisson process with constant rate $\lambda$ (occurrence of damage corresponds to "arrival of customers" in queuing theory). Whenever there is damage, a repair process starts, and the time it takes to repair the damage follows an exponential distribution with rate parameter $\mu$ (repair corresponds to "service" in queuing theory). Assume further that there is at most one repair process active at any time, implying that subsequent damages might accumulate, even if $\lambda << \mu$; i.e., there might be a "queue" of damage. This last assumption is made to simplify derivations but can be relaxed without changing the results qualitatively, i.e., the results in this section hold also for M/M/c queues. Since damage happen at random times, and take random amounts of time to repair, the accumulated damage, $Q(t)$ say, is a stochastic process on the non-negative integers ("queue length" in queuing theory). If $\lambda < \mu$ then $Q$ has a stationary distribution, and this distribution will play a central role in the following. In this M/M/1 model, the stationary distribution of $Q$ only depends on the ratio of damage to repair rate, $\rho \stackrel{\text{def}}{=} \lambda/\mu$ ("traffic intensity" in queuing theory), and takes the following simple form [14, p.62]:

$$\Pr(Q \geq k) \stackrel{\text{def}}{=} P(Q \geq k) = \rho^k.$$

Fig 1 illustrates samples of $Q(t)$ for three values of $\rho$ and the insets show the corresponding stationary distributions. It can be seen that as $\rho \to 1$, it becomes more probable that the accumulated damage ($Q(t)$) takes large values.

To turn this queuing model into a model of aging and death, two properties will be added. First, the rate of the repair process is made to decrease as a function of age (i.e., time), and, second, the risk of "death" will be linked to the accumulated damage $Q(t)$.

To keep derivations simple, the following form for the decrease of $\mu$ is assumed:

$$\mu(a) = \mu_0 - \beta a. \tag{1}$$

Here $\mu_0$ and $\beta$ are constants, with $\mu_0 > \lambda$, and $a$ denotes age. Accordingly, the rate of repair is decreasing linearly with age. Note that $\mu_0$ represents the initial repair capacity of the system, at age zero. If, as is the case for many human populations, the exponential dependence between mortality and age starts at an age $a_0 > 0$, Eq 1 is replaced by $\mu(a) = \mu_0 - \beta(a - a_0)^+$, where $(x)^+$ denotes the positive part of $x$.

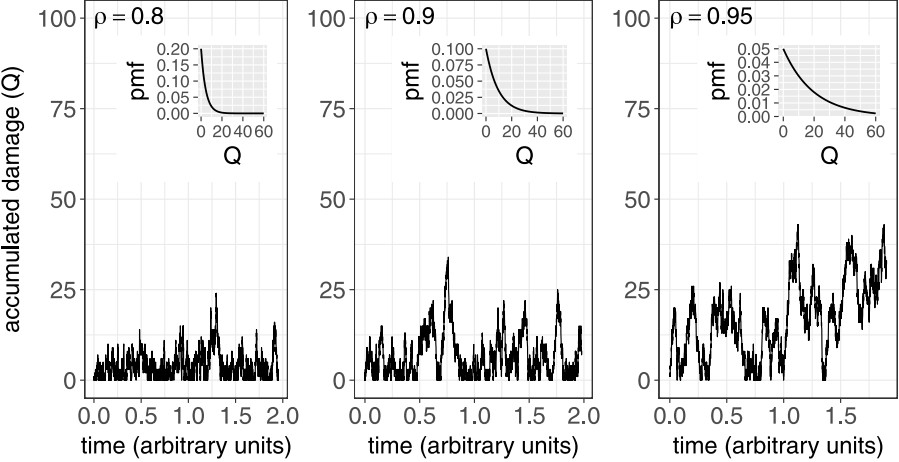

**Fig 1. Sample traces of $Q(t)$ for three different values of $\rho$ ($\lambda = 1000$).** Insets show the corresponding probability mass functions.

If $\mu$ is a function of age, then the distribution of $Q(t)$ will also depend on age, and when $\mu$ changes, it will take some time before the distribution of $Q$ conforms to the stationary distribution corresponding to the new value of $\rho$ (see Fig 4 for an illustration). In fact, the time it takes for the stationary distribution to be approached is a rapidly increasing function of $\rho$ [e.g., 15]. To accommodate this fact, it is further assumed that the rate of change of $\mu$ is slow compared to the time needed to approach the steady state distribution. This is achieved when the rate of damage is high compared to the rate of change of $\mu$, i.e., when $\lambda >> \beta$. With this assumption in place, the stationary distribution of $Q$ can be parameterized by age, $P(Q(a))$ as

$$P(Q(a) \geq k) = \rho(a)^k \overset{\text{def}}{=} \left( \frac{\lambda}{\mu(a)} \right)^k. \tag{2}$$

To link the accumulated damage to risk of dying, the latter is assumed proportional to the probability of the accumulated damage exceeding a fixed threshold $\theta$. In other words, the hazard of death at age $a$, $h(a)$ say, is given by

$$h(a) \propto P(Q(a) \geq \theta), \tag{3}$$

where the threshold, $\theta$, is a free parameter of the model system. In the S1 Appendix it is shown that similar behavior can be obtained with a soft threshold, a sigmoid function. The qualitative behavior of the system does not depend on the constant of proportionality, and it is assumed to be equal to one in the rest of this section. In applications, this constant is determined by the time units used. Given this form of the hazard function, it follows, from standard survival analysis theory, that the corresponding probability density function, $f(a)$ say, is given by

$$f(a) = \rho(a)^\theta \exp\left( -\int_{a_0}^a \rho(s)^\theta ds \right), \tag{4}$$

[e.g., 16, Ch.1]. Here $a_0 \geq 0$ to indicate that in real data the exponential dependence between age and mortality is often a restricted to a range of ages. Now we can state the

**Main result**. *Under these assumptions it follows that the rate of dying increases approximately exponentially with the age of the system.*

To see this, let $\Delta = \mu_0 - \lambda$ and use that

$$\rho(a) = \frac{1}{1 + \dfrac{\Delta - \beta a}{\lambda}}.$$

Insert this in the expression for the probability of $Q(a)$ (Eq 2) and take logarithms to yield

$$\log P(Q(a) \geq \theta) = -\theta \, \log\left( 1 + \frac{\Delta - \beta a}{\lambda} \right).$$

Taylor expansion of the logarithm gives,

$$\log P(Q(a) \geq \theta) = \theta \frac{\beta a}{\lambda} - \theta \frac{\Delta}{\lambda} + \text{ higher order terms}. \tag{5}$$

In the regime that is relevant here, say when $P(Q(a) \geq \theta) > 0.01$, the higher order terms in the above expression makes only a negligible contribution and can be ignored (see Fig 2 for an example). Thus, in this range, the logarithm of $P(Q)$ is very well approximated by a straight line. In other words, the probability that the accumulated damage exceed a given threshold $\theta$ is

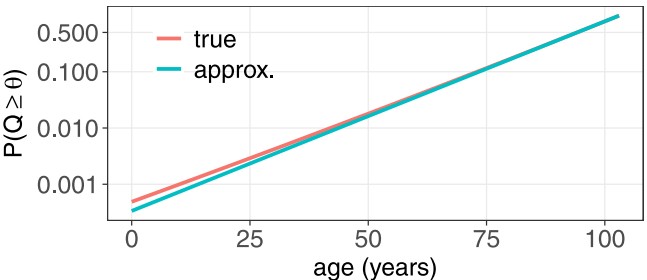

**Fig 2. Probability of $Q(a) \geq \theta$ as a function of age for a model with $\lambda = 500$, $\Delta = 50$, $\beta = 0.485/365$, all in units of per day, and $\theta = 80$.** True values from Eq 2 and approximation from Eq 6.

an exponential function of age. Indeed, if $c_1 = \exp(-\theta\Delta/\lambda)$, and $c_2 = \theta\beta/\lambda$, we have

$$P(Q \geq \theta) \simeq c_1 \exp(c_2 a), \qquad (6)$$

or equivalently $\log(P) \simeq \log(c_1) + c_2 a$. Fig 2 illustrates how the error in the approximation goes to zero as $P(Q \geq \theta) \to 1$.

Note that the assumption of a perfectly linear decrease of rate of repair (Eq 1) can be relaxed. The decrease does not need to be deterministic as long as the average decrease is roughly linear.

## Effects of changes in parameters

The model system has four parameters: $\mu_0$, $\lambda$, $\beta$, and $\theta$, and the way mortality rate depends on these can be obtained by taking partial derivatives of $P(Q)$ with respect to the parameters (e.g., using Eq 6). These dependencies are illustrated in Fig 3, which shows that changes in $\mu_0$ leads to a proportional change in the hazard rates, i.e., the hazard rate ratios between two systems that differ only in $\mu_0$ are constant. A change in the damage rate ($\lambda$) has similar effects (but with opposite sign), whereas a change in $\beta$ leads to hazard rate ratios that increase with age, and a change in $\theta$ to hazard rate ratios that decrease with age.

Since the relation between age and log mortality rate is almost linear in this model, it can be adequately be described by just two parameters (see Fig 2). This implies that the four parameters of the model cannot be uniquely fit to a single set of data. For example, if $\mu_0$, $\lambda$, and $\beta$ are all multiplied by the same constant, $P(Q)$, as given by Eq 6, will not change. However, the point is not primarily to fit the model to a single set of data, but to interpret changes in mortality rates between different contexts in terms of parameters with a clear interpretation.

## Damage accumulation modeled by a G/G/1 queue

To derive the approximate exponential dependence between age and mortality (Eq 6), it was assumed that damage happened according to a Poisson process and that the repair times were exponentially distributed. In this section it is shown that mortality rate is an approximately exponential function of age also when these assumptions are relaxed.

So, assume that damage occurs according to a stationary stochastic process, where the time intervals between damages are drawn from a distribution with mean value $1/\lambda$ and standard deviation $\sigma_D$. Assume further that repair times are drawn from a distribution with age-dependent mean value $1/\mu(a)$ and standard deviation $\sigma_R(a)$. Let $I_D$ and $I_R$ denote the square of the

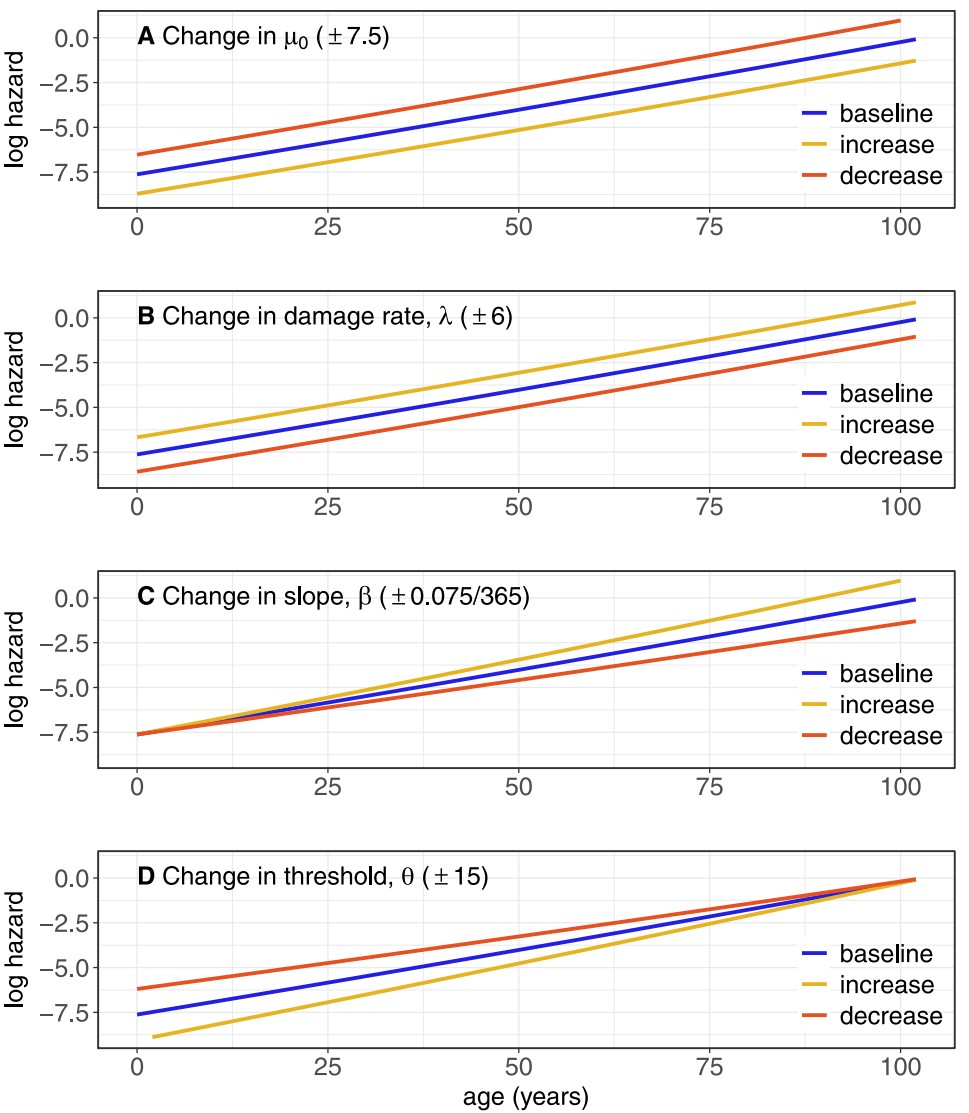

**Fig 3. Effects of changes in the parameters on $P(Q \geq \theta)$, using Eq 2 with $\lambda = 500$, $\mu_0 = 50$, $\beta = 0.485/365$, $\theta = 80$.**

coefficient of variation of these distributions, e.g., $I_D = (\lambda \sigma_D)^2$. When $\lambda/\mu \to 1$, it can be shown that the stationary distribution of accumulated damage, for a fixed age, $Q(a)$, is approximately exponential [e.g., 17, Ch. 8]. Indeed, in this case we have that

$$P(Q(a) \geq \theta) \simeq \exp\left(-2\theta \frac{\mu(a) - \lambda}{I_R \mu(a) + I_D \lambda}\right). \tag{7}$$

To show that this expression entails an approximate exponential dependence between age and mortality rate, assume that $I_D$ does not depend (strongly) on age and use that

$$
\begin{aligned}
\frac{\mu(a) - \lambda}{I_R \mu(a) + I_D \lambda} &= \frac{\Delta - \beta a}{I_R(\Delta - \beta a) + (I_R + I_D)\lambda} \\
&= \frac{\Delta - \beta a}{(I_R + I_D)\lambda} - \frac{I_R(\Delta - \beta a)^2}{I_R(I_R + I_D)\lambda(\Delta - \beta a) + ((I_R + I_D)\lambda)^2}.
\end{aligned}
$$

Next, we show that the second term on the right hand side in the above expression is smaller than the first term and can be ignored when $(\mu - \lambda) \to 0$. To do this, let $r$ denote the ratio between the second and the first term, i.e.,

$$|r| = \frac{I_R(\Delta - \beta a)}{I_R(\Delta - \beta a) + (I_R + I_D)\lambda} . \tag{8}$$

The absolute value used in this expression follows from the assumption that $(\Delta - \beta a) \geq 0$. Eq 8 implies that $|r| < (\Delta - \beta a)/\lambda$, showing that $|r| \to 0$, when $(\Delta - \beta a) \to 0$, demonstrating that the contribution of the second term can be ignored when $(\mu(a) - \lambda)$, and hence $(\Delta - \beta a)$ is small. Consequently, in this regime, we can approximate Eq 7 by

$$P(Q(a) \geq \theta) \simeq \exp\left(-2\theta \frac{\Delta - \beta a}{(I_R + I_D)\lambda}\right), \tag{9}$$

which shows that the probability of damage exceeding a given threshold is approximately an exponential function of age. Note that Eq 9 express the same dependence between $P(Q)$ and the model parameters as does Eq 6. Furthermore, since the coefficients of variation is 1 for both damage and repair distribution in the M/M/1 model above, the approximation in Eq 9 becomes, in this case, identical to that in Eq 6.

To verify that the theoretical approximations derived in this section accurately describe the behavior of the model systems, direct numerical simulations of two systems were performed and the results of these were compared to the theoretical approximations (see S1 Appendix). Fig A2 in S1 Fig shows that the approximation of Eq 9, accurately captures the dependence between age and hazard rate, in particular as the rate of damage approaches the rate of repair.

To sum up, there are two factors explaining why there is an exponential dependence between age and mortality rate in these models. The first has to do with the shape of the stationary distributions of $Q(t)$, geometric in the M/M/1 case, and exponential in the general case. This is a property of the queuing models used to describe the damage accumulation. The second factor is that the model systems are working in a regime where $\mu(a) \to \lambda$, i.e., where the repair rate is approaching the damage rate. In this regime, the stationary distributions have an approximately linear dependence on age. At this abstract level it is almost self evident that real biological systems must also be operating in this regime as they age. Indeed, if the rate of repair would continue to be much higher than the rate of damage, there would not be any accumulation of damage, and consequently, no aging.

## Behavior of models when λ > μ

To derive analytical results for the model systems it has, so far, been tacitly assumed that $\lambda < \mu$, i.e., that the rate of repair is higher than the rate of damage. When this condition fails, there is no stationary distribution and the probability of $Q$ exceeding any threshold becomes equal to one. In other words, damage will accumulate without bounds and the probability that $Q$ will exceed a fixed threshold $\theta$ becomes independent of age. Since stationary theory cannot be used to derive analytical expressions in this regime, numerical simulations (see S1 Appendix) were used to illustrate the behavior. Fig 4 shows results for one set of parameters. It is clear that as $\rho$ gets close to 1, the simulated data starts deviating slightly from the theoretical prediction. This is a consequence of that it takes more time to reach the stationary distribution when $\rho \to 1$. It is also clear from Fig 4 that the exponential increase in hazard rate, is seemingly completely gone when $\rho \geq 1$, and the hazard of death becomes, to a first approximation, independent of $\rho$, and hence age.

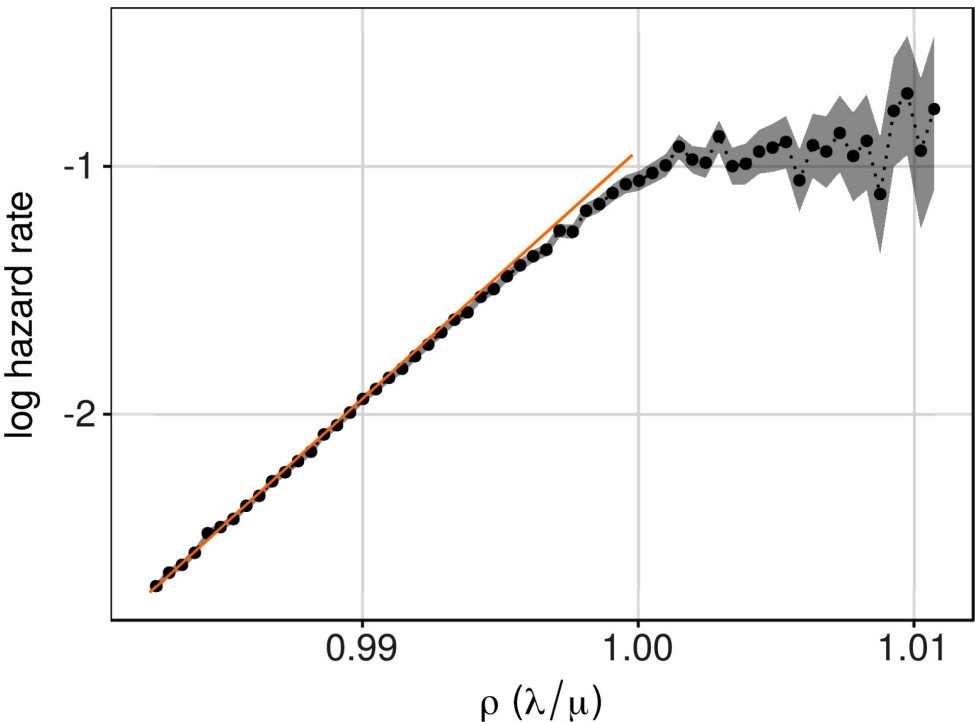

**Fig 4. Hazard rates as a function of $\rho$, i.e., $(\lambda/\mu)$, close to 1.** The dashed region represents 95% confidence intervals (from a Poisson approximation), and the orange line represents predictions from the theory (Eq 6). Model parameters were: $\lambda = 540$, $\mu_0 = 550$, $\beta = 0.52/365$, $\theta = 100$, and $4.0 \times 10^5$ realizations of the systems were run.

## Heterogeneity

The analytical results considered so far concerns the behavior of a large number of identical units. In populations of biological organisms it is likely to exist differences between organisms that affect their longevity. For example, females tend to outlive males in many animal species, and various experimental manipulations have been shown to systematically alter longevity in animal models [e.g., 2]. Such heterogeneity can easily be incorporated into the model systems described here. For example, if the initial level or repair capacity, $\mu_0$, is a random variable such that $\log(\mu_0)$ has a suitable gamma distribution, the resulting model would generate hazard rates, as a function of age, almost identical to the, so-called, gamma-Gompertz model, in which heterogeneity is modeled by a proportional scaling of the hazards [e.g., 18, 19]. A perhaps more principled choice would be to let the initial repair capacity be normally distributed; this would then instead imply a log-normal proportional frailty model. Another possibility is to let the damage rate $\lambda$ vary between units. Analytical results for model systems where parameters vary according to a normal distribution have not been derived yet, but using numerical simulations, some consequences of heterogeneity is studied below.

A prominent feature of the gamma-Gompertz frailty model is that the hazard rates grow less fast at older ages. The reason for this is that those that survive to older ages tend to have a lower hazard than the average of the population [e.g., 20]. To show that this effect is also observed when heterogeneity is introduced in the model systems considered here, numerical simulations were run in which the initial repair rate $\mu_0$ was allowed to vary between units. In Fig 5, hazard rates from these simulations are compared to the hazard rates from a model

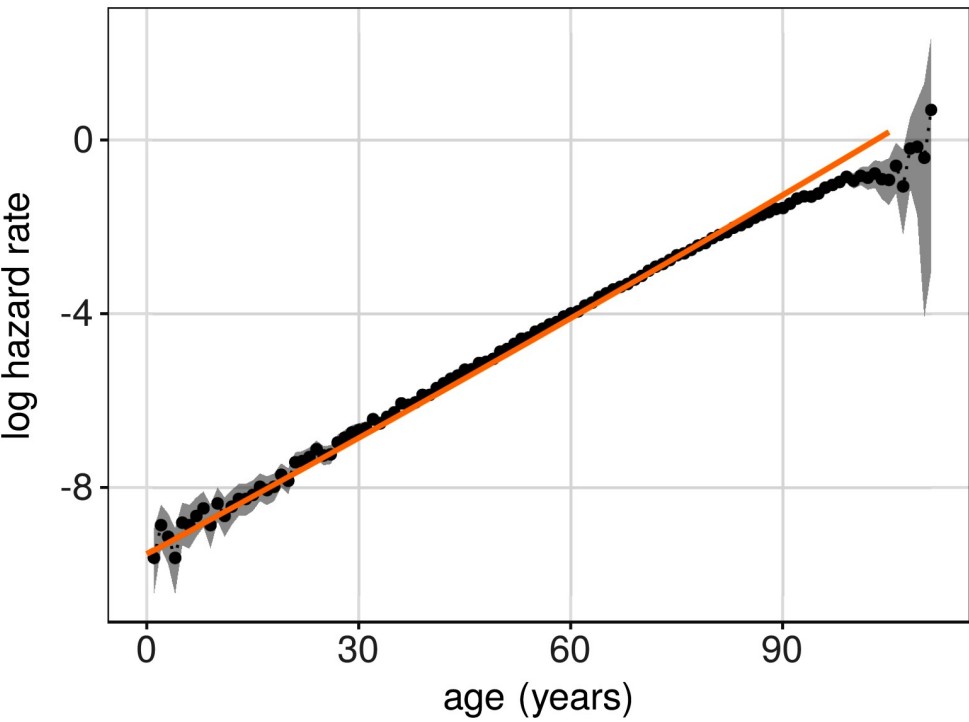

**Fig 5. Estimated hazard rates (points) for a model system with heterogeneity in $\mu_0$.** The gray region represents 95% confidence intervals (from a Poisson approximation), and the orange line represents hazard rates for a model system without heterogeneity. Heterogeneity was modeled by sampling values of $\mu_0$ from a normal distribution with mean 550 and variance 4. Other model parameters were: $\lambda = 500$, $\beta = 0.485/365$, $\theta = 100$. Results are based on $1.2 \times 10^5$ model realizations.

without heterogeneity. The hazard rates are very similar until age 75 after which the rates in the population with heterogeneity show a slower increase. Similar results are obtained by allowing $\lambda$ to vary (not shown).

Heterogeneity in $\mu_0$ or $\lambda$ makes sense in terms of interpretation. Differences in the repair capacity between different individuals might be a result of genetic differences, and differences in rate of damage might be caused by living conditions and/or lifestyle (e.g., smoking, drinking alcohol). The effects of heterogeneity in other parameters is left for a separate communication.

## As a phenomenological model of real data

In this section the model where damage accumulation is governed by an M/M/1 queue is fit to historical mortality data from Sweden. The purpose is to illustrate how the system can be used as a phenomenological model of real data, and thereby to aid in the interpretation. A more detailed analysis and discussion of results will be communicated separately.

### The data

Mortality data for persons born in Sweden were obtained from the "Swedish Book of Deaths" issued by the The Federation of Swedish Genealogical Societies. This is a database compiled from a range of sources and contains information on time and place of births and deaths for persons that have died in Sweden since 1860. The coverage is almost complete.

Birth cohorts were formed for the two years 1885 and 1905, for men and women separately (men 1885: $N$ = 58044, 1905: $N$ = 64602; women 1885: $N$ = 57873), 1905: $N$ = 63284). These birth years were selected since it is known that the mortality rates for the 1905 cohort are significantly lower than the rates of the 1885 cohort. Hazard rates were estimated with a resolution of 1 year. To find the models that best fitted these four sets of data, a two-step procedure was used. First, a linear least squares fit was made to the log hazards for each cohort separately. The linear fits were restricted to an age range of 55-99 in order to capture the interval in which there is an exponential increase of mortality in these data, and to exclude older ages where the uncertainties of the hazard-rate estimates are large. The parameters from the least squares fits (two parameters per birth year and sex) were subsequently used to find the corresponding parameters of the model systems. This was accomplished by solving numerically the following system of equations for $\lambda$ and $\theta$:

$$c_o + c_1 55 = \theta \, \log \left( \frac{\lambda}{\mu_0 - \beta 55} \right)$$

$$c_o + c_1 90 = \theta \, \log \left( \frac{\lambda}{\mu_0 - \beta 90} \right),$$

where $c_0$ and $c_1$ are the intercept and slope from the linear least squares fit to the log hazards. The other two model parameters, $\mu_0$ and $\beta$, were held fixed at 550 and 0.485/365. All computations were made in R [21].

Fig 6 shows hazard rates for the two birth years, for men and women separately. As is well known, the mortality rates for women dropped for almost all ages (the spikes around 14 years of age in the 1905 data, and 34 in the 1885 data, are attributed to the Spanish flu which hit Sweden in the fall of 1918). For men, mortality rates reduced for ages under 55, but did not change much for older ages. The lower panels show the data restricted to ages $\geq 55$. The exponential increase in mortality rates is captured by the model (solid lines). The parameters that best describe these data show that the substantial decrease in hazard rates for women, after age 55, is accounted for by an increase in threshold $\theta$; the rate of damage, $\lambda$, remained almost exactly the same in the two cohorts (Table 1). This increase in tolerance to damage is tentatively attributed to the the improvements in health care that occurred during the second half of the 20th century, e.g., in availability of antibiotics [22], as these improvements must have benefited those born 1905 more than those born 1885. An explanation for why mortality among men did not decrease to the same extent, among those aged 55 and older, might be found in the different rates in cigarette smoking in these two cohorts, and consequences thereof. Cigarette smoking became widespread among Swedish men in the 1940s, and were picked up by the two birth cohorts to a different extent [23, 24]. For example, according to a 1955 national survey on smoking and tobacco use, 41% of men aged 50-65 (i.e., including the 1905 birth cohort) were smoking cigarettes, whereas only 20% of men older than 65 did (i.e., including the 1885 birth cohort). Smoking among women in the corresponding age groups were 20 and 11% respectively [23]. A more careful investigation of these claims is left for a separate contribution. However, the fact that the model could account for the substantial change in mortality seen in women by a change in one sensible parameter is encouraging for further application to real mortality data.

## Discussion

The model systems introduced herein accumulate damage and die with a rate that depends on the age of the system. If the decrease in the rate of repair is linear with age, mortality will increase exponentially, similarly to what is observed in many biological systems. It was shown

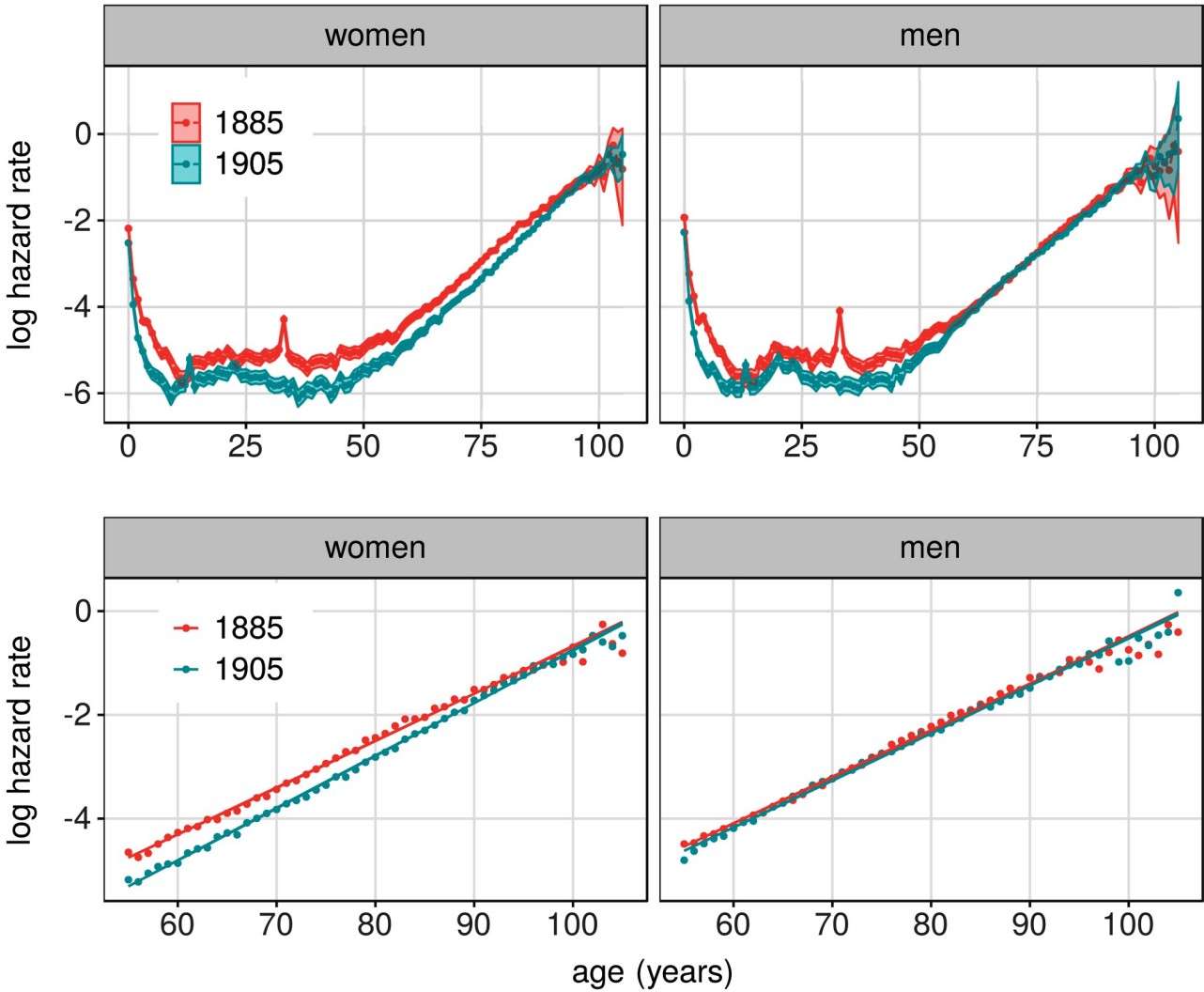

**Fig 6. Hazard rates estimated from mortality data from Sweden.** Top panels show estimated mortality rates for men and women born 1885 and 1905. Shaded areas show extent of 95% confidence intervals of these estimates. Lower panels show the same data, but restricted to ages ≥ 55. Solid lines in lower panels show hazard rates from the model fits.

that the exponential increase is independent of the interval distributions governing the damage and repair processes, demonstrating that this is a robust feature of these systems. The exponential dependence between age and mortality rates in these models is valid in a regime where the rate of repair becomes similar to the rate of damage as the systems age. Since aging, in both

**Table 1. Parameters of the models that best fit the data.** The value of the other parameters were held fixed at: $\mu_0 = 550, \beta = 0.485/365$. The $R^2$ column shows the $R^2$ value calculated over the age range 55 to 99.

| Birth year | Sex | $\lambda$ | $\theta$ | $R^2$ |
|---|---|---|---|---|
| 1885 | W | 498.1 | 96.1 | 0.997 |
| 1905 | W | 498.1 | 107.3 | 0.995 |
| 1885 | M | 499.0 | 95.5 | 0.998 |
| 1905 | M | 498.9 | 96.7 | 0.996 |

animals and humans, is associated with accumulation of damage [e.g., 2, 8–11], it is possible that the exponential relation between age and mortality also in these cases results from a similar balance of damage and repair.

In the current form, these model systems have four parameters, all with clear interpretations: rate of damage accumulation, initial rate of repair, rate of decrease of repair rate, and a threshold beyond which the accumulated damage can cause the system to die. Although it it possible to express the dynamics of the model with fewer parameters, such reparameterization would make the interpretation of the model harder. An important finding is that the way these parameters affect the relationship between age and mortality is independent of the details of the accumulation process. In other words, it is a generic feature of these systems. For example, an increase in the damage rate will lead to a proportional increase of the mortality rate (see Fig 3) in *all* models of this type. This generic property suggests that similar parameter dependencies might hold also in other (biological) systems. It was, for example, shown that the decrease in mortality rates among Swedish women born 1905, compared to those born 1885, could be well accounted for by an increase in just one parameter, $\theta$, the threshold. This example also illustrates how model systems, where parameters have an intrinsic meaning, can be used to generate hypotheses about generative mechanisms, something that might be harder to do with descriptive (statistical) models of the data.

Apart from the exponential dependence between aging and mortality, human mortality data often show other regularities that a model should arguably be able to reproduce. One common finding is that the exponential increase in hazard rates levels off at old age, i.e., the increase in mortality with age, say after 95, becomes less fast (but this is not a universal finding, see [25]). In the models introduced here, such leveling off is achieved naturally when the rate of damage is close to, or surpasses, the rate of repair (Fig 4). Heterogeneity in the initial level of repair provides another way in which this behavior can be generated (Fig 5). Another common feature of human mortality data is a negative correlation between the intercept and slope when the log hazard is modeled as a linear function of age (i.e., a straight line), the so-called Gompertz model [26]. In the model systems considered here, such dependency is a consequence of changes in model parameters that influence both the slope and intercept of the log hazard, see Eq 6. For example, in populations were mortality rates change through a change in the threshold, there will be a strong negative correlation between the intercept and slopes in the corresponding Gompertz models. Consequently, these models give a functional explanation to why such a dependency might be found in real populations [c.f., 27].

Below I first discuss some ways in which these models can be extended. Subsequently, two conjectures concering the relation between parameters of the models and real data are put forth. Hopefully this makes the connections to real systems more clear. The paper concludes with a discussion of some related work.

## Extensions

There are many ways in which the family of damage-accumulation models considered here can be extended. One immediate extension would be to consider the accumulated damage, not as an integer-valued random variable, but a continuous ditto. In fact, the diffusion approximation underlying Eq 7, is an instance of a reflected Brownian motion [17]. This means that reflected (at zero) Brownian motion with a drift $\mu$ that is increasing as a function of age (initially $\mu < 0$) provides an alternative model of damage accumulation. If $\mu$ is increasing linearly with age, this model will also lead to exponentially increasing mortality rates. Discrete models were used in this work because the parameters in these models are more intuitive, however, a continuous formulation might be preferable for analytical work.

Another way that the models could be extended would be to allow the rate of repair to depend on the accumulated damage. Indeed, in real systems, it is likely that the decrease in the efficiency of repair with age is itself a consequence of damage accumulation. If the rate of mortality would show the same dependency on age in this case is not known, but it is likely that it would be approximately true for certain types of dependencies.

Yet another extension concerns damage and repair processes that are non-stationary. If changes in the properties of these processes are slow, compared to the average interval duration, the results will not change qualitatively [e.g., 28]. Indeed, the linear decrease in $\mu$ with age, used in the models studied herein, is exactly such a slow change. If a model system with faster changes in damage and repair would still show an exponential dependence between mortality and age, or not, likely depends on the details of these changes.

To use the model systems studied here to model the mechanisms of aging in a particular organism would require additional steps. The damage and repair in the model systems should then be put in correspondence to some processes known to occur in the real system. In biological systems accumulation of damage happens at many different levels of organization (e.g., organelles, cells, functional modules, organs, etc), at different time-scales, and probably with different rates; and a biologically realistic model should be explicit about which levels are modeled and how different levels interact. Future work will explore how hierarchical combination of simple queuing models can be used to build more biologically realistic models of aging.

## Two conjectures

The main aim of this work was to show that an exponential increase in mortality rates is a generic consequence of damage accumulation in an abstract model system. Hopefully this theoretical work can contribute to more realistic models of biological mechanisms of aging. However, even at this abstract level it is possible to establish some tentative connections to real mortality data and therefore to mechanisms of aging. To this aim I will propose two testable conjectures regarding how established empirical findings relate to changes in model parameters.

Two of the four model parameters are related to external conditions of life: the rate of damage accumulation $\lambda$, and the threshold $\theta$. It was already shown that the decrease in mortality among women born in Sweden could be fully accounted for by an increase in the threshold (Table 1). Exactly how different interventions translate to changes in the threshold is not known, but an improvement of, and increased access to, health care should correspond to a higher threshold. Thus, the increase in threshold found here is consistent with the marked improvements in health care in Sweden during the 20th century. The damage rate $\lambda$ was considered constant in the models, however, in real biological systems it is likely that the rate of damage varies due to variations in external conditions. The first conjecture relates to this parameter:

C1 The observed seasonal variations in mortality [e.g., 29, 30] can be accounted for by seasonal variations in the rate of damage accumulation.

A quantitative test of this conjecture is under way and will be communicated separately.

The other two model parameters, initial rate of repair, $\mu_0$, and rate of decay of the repair, $\beta$, supposedly reflect intrinsic properties of the organisms. For example, these are the parameters that would account for genetic differences in longevity in different populations. The amount of variation of longevity that can be attributed to genetic factors (i.e., heritability) has been shown to be in the order of 25 percent [31, 32]. If changes in $\mu_0$ or in $\beta$ best account for these inheritable differences in longevity is not known. However, by comparing mortality rates of a population selected, on the basis of family data, to have longer life span, to rates from a

population selected to have shorter life span, this question can be investigated empirically. Indeed, the second conjecture is:

C2 The differences in mortality rates between two populations selected to have long and short life spans, respectively, can be accounted for by a change in $\mu_0$.

This conjecture can be tested on data from twin studies, for example by comparing mortality rates of those who had a twin sibling that died after reaching a given age (80 years say) with those who's twin siblings died before this age.

## Relations to previous work

A number of model systems previously described also generate mortality rates that increase more-or-less exponentially with age [e.g., 26, 33–37]. From a mathematical point of view, several of these models are similar to the ones presented here in that they also model a state variable that changes due to stochastic perturbations and the system dies when the state variable exceeds some given level. In fact, one of the models considered by Sacher and Trucco [33] is a one-dimensional Brownian motion with two absorbing boundaries, very similar, in kind, to the reflected Brownian motion mentioned above. However, the continuous formulation of that, and other, models makes it harder to interpret the model parameters in an intuitive way. For example, in the models that are formulated in terms of diffusion equations [33, 35] there is no natural connection between the drift term and the diffusion term. In the queuing framework suggested here, these terms are intrinsically connected since they are derived from a common underlying model. Furthermore, the discrete queuing-based models considered here arguably provide a simpler explanation for the exponential relation between age and mortality rates.

All these models, both previously described and the ones introduced in this work, are abstract in the sense of not being committed to any real system, and therefore it is not very meaningful to compare them in terms of "best fitting the data" (most of them will fit mortality data equally well), and the choice of model should instead be dictated by other criteria such as interpretability and usefulness as a tool to understand and explain data. I hope that the relative simplicity of the model systems introduced here, together with the ease at which they can be analyzed, understood, and fit to data, are convincing arguments for them to become useful complements to existing models.

In the present model there is a state variable, the accumulated damage, $Q(t)$, the value of which determines the probability of dying. What this state variable might correspond to in a real aging organism remains to be determined. However, it is clear that it would have to be a summary measure somehow capturing the state of the organism at a given time point. Interestingly, there is a substantial body of work from Mitnitski, Rockwood and colleagues highly related to this issue. In a series of papers they have introduced and evaluated what they refer to as a "frailty index", a composite measure of frailty, computed from a large number of frailty indicators [e.g., 10, 38, 39]. They have shown that the frailty index predicts mortality with higher accuracy than age alone, and have suggested that it could serve as a proxy for biological age Mitnitski Rockwood2015. In modeling work from the same group, they have used a framework very similar to the one proposed here to model how the frailty index evolves over time [10, 40]. In particular, they use the average queue length to represent the frailty index. Future work should explore the connection between $Q(t)$ and frailty index in more detail.

## Supporting information

**S1 Appendix. Comparison with numerical simulations.**
(PDF)

**S1 File. Data and code for Fig 6.**
(ZIP)

**S1 Fig.**
(ZIP)

## Acknowledgments

I thank Ylva Almquist for helping me acquiring the Swedish mortality data and Sveriges Släkt-forskarförbund for letting me use data from Dödboken.

## Author Contributions

**Conceptualization:** Anders Ledberg.

**Data curation:** Anders Ledberg.

**Formal analysis:** Anders Ledberg.

**Funding acquisition:** Anders Ledberg.

**Investigation:** Anders Ledberg.

**Methodology:** Anders Ledberg.

**Project administration:** Anders Ledberg.

**Resources:** Anders Ledberg.

**Software:** Anders Ledberg.

**Validation:** Anders Ledberg.

**Visualization:** Anders Ledberg.

**Writing – original draft:** Anders Ledberg.

**Writing – review & editing:** Anders Ledberg.

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
