## [Decision Letter · Decision Letter 0]

20 Apr 2020

PONE-D-20-04967

Exponential increase in mortality with age is a generic property of a simple model system of damage accumulation and death

PLOS ONE

Dear Dr. Ledberg,

Thank you for submitting your manuscript to PLOS ONE. After careful consideration, we feel that it has merit but does not fully meet PLOS ONE’s publication criteria as it currently stands. Therefore, we invite you to submit a revised version of the manuscript that addresses the points raised during the review process. As suggested by a reviewer we would like you to consider  and discuss a practical application of the knowledge gained by your modelling approach. We would appreciate receiving your revised manuscript by Jun 04 2020 11:59PM. To enhance the reproducibility of your results, we recommend that if applicable you deposit your laboratory protocols in protocols.io, where a protocol can be assigned its own identifier (DOI) such that it can be cited independently in the future. For instructions see: http://journals.plos.org/plosone/s/submission-guidelines#loc-laboratory-protocols

We look forward to receiving your revised manuscript.

Kind regards,

Miguel Angel Luque-Fernandez

Academic Editor

PLOS ONE

Journal Requirements:

1. Thank you for including your funding statement; "This research was partly financed by a grant from SiS with number 2.6.1-1134-2017."

3. We note you have included a table to which you do not refer in the text of your manuscript. Please ensure that you refer to Table 1 in your text; if accepted, production will need this reference to link the reader to the Table.

Reviewers' comments:

Reviewer's Responses to Questions

**Comments to the Author**

1. Is the manuscript technically sound, and do the data support the conclusions?

Reviewer #1: Yes

Reviewer #2: Partly

2. Has the statistical analysis been performed appropriately and rigorously? 

Reviewer #1: Yes

Reviewer #2: I Don't Know

3. Have the authors made all data underlying the findings in their manuscript fully available?

Reviewer #1: Yes

Reviewer #2: Yes

4. Is the manuscript presented in an intelligible fashion and written in standard English?

Reviewer #1: Yes

Reviewer #2: No

5. Review Comments to the Author

Reviewer #1: I like the idea that the Gompertz law reflects a generic property of a rather simple model of damage accumulation leading to death. I also like the idea to include the rate of repair in the model and do so in the framework of the stochastic queuing theory. I particularly like the idea that damage can be modeled by the length of the queue (look at some references below). However, I have several essential comments that if accounted might increase the quality of the manuscript.

First, several critical references are missed. (1) Assessing biological aging: the origin of deficit accumulation. Mitnitski A, Song X, Rockwood K. Biogerontology. 2013 Dec;14(6):709-17. doi: 10.1007/s10522-013-9446-3. Epub 2013 Jul 17. (2) Aging as a process of deficit accumulation: its utility and origin. Mitnitski A, Rockwood K. Interdiscip Top Gerontol. 2015;40:85-98. doi: 10.1159/000364933. Epub 2014 Oct 13. Review. The model deals with the mean values (in difference with your approach) but I think that what you suggested is quite important although it will be even more important to make it more transparent for the geroscience audience. Perhaps it will be easier for theoretical biologists to understand the matter.

Another useful reference (although not directly related to the model but relevant for the context) is Kirkwood TB, Deciphering death: a commentary on Gompertz (1825) 'On the nature of the function expressive of the law of human mortality, and on a new mode of determining the value of life contingencies'. Philos Trans R Soc Lond B Biol Sci. 2015 Apr 19;370(1666). pii: 20140379. doi: 10.1098/rstb.2014.0379.

Perhaps you would update the section 7.2 tasking into account the references above.

Second, it is not clear to me how essential is the introduction of the threshold of the equation for death, although you introduced a soft threshold in appendix B. Wouldn’t be enough to use the hazard as a function of damage, not age?

Finally, you payed considerable attention to the conditions and the role of the stationary distribution in describing the damage accumulation. Indeed, in the models, the very existence of such distribution depends on the traffic intensity which has to be less than 1. I would like to attract your attention to the idea of using a quasi-stationary distribution- when the parameter changes slowly (the aging process is slow comparing to the fast process of relaxation (repair) – a way usually employed in such situations in many applications in physics and engineering. Actually, you have mentioned that in one way or other and that is why numeric simulation is important.

Reviewer #2: Honestly, this is a very difficult paper for me to review. Of course I'm intimately familiar with the underlying premise of the paper, which is that the dying out process of living things like humans, follows a characteristic path that was identified almost 200 years ago. You use mathematical modeling to address four basic principles: mortality risk is associated with the rate of damaging accumulation in the body (one of the basic ideas in our field that's been around for decades); living things evolved mechanisms for addressing this accumulated damage -- which is to repair it with some level of efficiency (repair can be to DNA or other systems); there is some initial rate of repair parameter that can be identified; and of course the repair mechanism itself ages or becomes less efficient as time passes or as the organism grows older. The idea itself is not new, but what you did was develop a purely mathematical model describing these four parameters, and then applied this to data from Sweden -- looking at two different birth cohorts to illustrate its utility. Truth be told, the math is beyond my comprehension, so I can't even comment on it -- there are other mathematical demographers that are more well suited to addressing the issue of whether the math is correct. What caught my attention here was the use of a purely mathematical model to describe an inherently biological phenomenon. This is my personal bias, and I see the utility in creating a mathematical model of the dying out process that avoids any direct connection to the mechanisms underlying the processes being observed, but then I ask myself what I've learned from this exercise. The answer is that math can, at times, be used to simplify and help understand an incredibly complex phenomenon such as biological aging. There are two things that I would have liked to have seen done with this, assuming that the math is right of course. The first is to place this mathematical model within the context of the evolutionary theory of senescence. In this regard, one would be trying to understand WHY different species have different repair capacities, and why different repair capacities and accumulated damage can be exhibited in cohorts of people born in different eras. The answer may be found in a single word -- reproduction. If duration of life is calibrated to elements of the reproductive life history of the species, then the accumulation of damage and the efficiency of the repair mechanisms and the aging of the repair mechanisms, are all calibrated life history traits that are fundamentally driven by the level of hostility in the environment. What I'm saying here is that the author is missing the big picture. Can papers like this be published without the big picture? Sure. But the problem is that it does not do much to advance the field of aging science. Again, this is my personal bias, and I don't think scientists should be penalized for not knowing the big picture when their primary goal is to simplify the big picture in the form of mathematical equations. The second thing I would like to have seen is to have you speculate on how these systems might be artificially manipulated by aging science. Researchers are trying to find a way to intervene in these processes, and the question that might be answered by this model is which of the various mechanisms in operation that influence duration of life, would be the best targets to intervene. Should we focus on the rate of repair; the aging of the repair mechanisms; addressing non-stochastic aging related events; what exactly does this tell us about what might work best? What I'm looking for here is a practical application of the knowledge gained by models like this.

Without a link to underlying mechanisms driving the entire process of aging, and without some use case, this appears as little more than an effort to generate equations to describe a biological phenomenon. Once again, there is a place for this in science of course, so I'm just letting you know what my personal bias is here. Published in its current form, the ideas will just vaporize into thin air as a mathematical exercise.

6. PLOS authors have the option to publish the peer review history of their article (what does this mean?). If published, this will include your full peer review and any attached files.

Reviewer #1: Yes: Dr Arnold Mitnitski

Reviewer #2: No

---

## [Author Response · Author response to Decision Letter 0]

30 Apr 2020

The response to the reviewers has been uploaded as a separate file.

---

## [Decision Letter · Decision Letter 1]

5 May 2020

Exponential increase in mortality with age is a generic property of a simple model system of damage accumulation and death

PONE-D-20-04967R1

Dear Dr. Ledberg,

We are pleased to inform you that your manuscript has been judged scientifically suitable for publication and will be formally accepted for publication once it complies with all outstanding technical requirements.

With kind regards,

Miguel Angel Luque-Fernandez

Academic Editor

PLOS ONE

Additional Editor Comments (optional):

Reviewers' comments:

Reviewer's Responses to Questions

**Comments to the Author**

1. If the authors have adequately addressed your comments raised in a previous round of review and you feel that this manuscript is now acceptable for publication, you may indicate that here to bypass the “Comments to the Author” section, enter your conflict of interest statement in the “Confidential to Editor” section, and submit your "Accept" recommendation.

Reviewer #1: All comments have been addressed

Reviewer #2: (No Response)

2. Is the manuscript technically sound, and do the data support the conclusions?

Reviewer #1: Yes

Reviewer #2: Yes

3. Has the statistical analysis been performed appropriately and rigorously? 

Reviewer #1: Yes

Reviewer #2: I Don't Know

4. Have the authors made all data underlying the findings in their manuscript fully available?

Reviewer #1: Yes

Reviewer #2: Yes

5. Is the manuscript presented in an intelligible fashion and written in standard English?

Reviewer #1: Yes

Reviewer #2: Yes

6. Review Comments to the Author

Reviewer #1: Thank you for accounting for my comments/ concerns. I believe that the manuscript improved sufficiently. I generally like it and see potentials of further investigations.

Reviewer #2: I have no additional suggested changes to the manuscript, although I would encourage the author to familiarize himself with the literature on the evolutionary biology of aging. I believe he will find this particularly enlightening given that the actuarial and biological sciences addressed this same issue roughly 125 years apart, and the only reason evolutionary biologists didn't know about the work of Gompertz was because they didn't bother to look for it. Once you read the works of Medawar and Williams, I believe you'll be shocked at how the underlying biological mechanisms of what you're observing mathematically, were identified more than 70 years ago.

7. PLOS authors have the option to publish the peer review history of their article (what does this mean?). If published, this will include your full peer review and any attached files.

Reviewer #1: Yes: Dr Arnold Mitnitski

Reviewer #2: No

---

## [Editor Report · Acceptance letter]

14 May 2020

PONE-D-20-04967R1 

Exponential increase in mortality with age is a generic property of a simple model system of damage accumulation and death 

Dear Dr. Ledberg:

I am pleased to inform you that your manuscript has been deemed suitable for publication in PLOS ONE. Congratulations! Your manuscript is now with our production department. 

With kind regards,

on behalf of

Dr. Miguel Angel Luque-Fernandez 

Academic Editor

PLOS ONE